# N-terminomics profiling of host proteins targeted by excretory-secretory proteases of the nematode *Angiostrongylus vasorum* identifies points of interaction with canine coagulation and complement cascade

Nina Germitsch[1,2¤a], Tobias Kockmann[3], Manuela Schnyder[1]*, Lucienne Tritten[1¤b]*

**1** Institute of Parasitology, Vetsuisse Faculty, University of Zurich, Zurich, Switzerland, **2** Graduate School for Cellular and Biomedical Sciences, University of Bern, Bern, Switzerland, **3** Functional Genomics Center Zurich, Swiss Federal Institute of Technology Zurich (ETH Zurich)/ University of Zurich, Zurich, Switzerland

¤a Current address: Pathology and Microbiology, Atlantic Veterinary College, University of Prince Edward Island, Charlottetown, Canada
¤b Current address: Institute of Parasitology, McGill University, Sainte-Anne-de-Bellevue, Canada
* lucienne.tritten@mcgill.ca (LT); manuela.schnyder@uzh.ch (MS)

**Data Availability Statement:** The generated mass spectrometry data have been deposited to the

## Abstract

The cardiopulmonary nematode *Angiostrongylus vasorum* can cause severe disease in dogs, including coagulopathies manifesting with bleeding. We analysed *A. vasorum* excretory/secretory protein (ESP)-treated dog plasma and serum by N-terminome analysis using Terminal Amine Isotopic Labelling of Substrates (TAILS) to identify cleaved host substrates. In plasma and serum samples 430 and 475 dog proteins were identified, respectively. A total of eight dog proteins were significantly cleaved at higher levels upon exposure to *A. vasorum* ESP: of these, three were coagulation factors (factor II, V and IX) and three were complement proteins (complement C3, C4-A and C5). Comparison with human motif sequence orthologues revealed known cleavage sites in coagulation factor IX and II (prothrombin). These and further identified cleavage sites suggest direct or indirect activation or proteolysis of complement and coagulation components through *A. vasorum* ESP, which contains several proteases. Further studies are needed to validate their substrate specificity.

## Introduction

Dogs infected with the cardiopulmonary nematode *Angiostrongylus vasorum* can suffer severe disease. Around one third of infected dogs manifest internal or open bleeding [1, 2], which may correlate with hypocoagulation and hyperfibrinolysis [3] and often ends in fatal outcome. Observed coagulopathies have been hypothesised to be due to a number of coagulation abnormalities including disseminated intravascular coagulation (DIC), up- or downregulation of anti- or procoagulant factor activity, thrombocytopenia, and others [1, 4, 5].

ProteomeXchange Consortium via the PRIDE (https://www.ebi.ac.uk/pride/) partner repository (dataset identifier PXD056659).

**Funding:** We would like to thank the University of Zurich ('Forschungskredit' FK-20-060), Switzerland, and Bayer Vital GmbH, Business Unit Animal Health, Germany for financial support to N. G. during her PhD studies. The funders had no role in study design, data collection and analysis, decision to publish, or preparation of the manuscript.

**Competing interests:** he authors have declared that no competing interests exist.

*Angiostrongylus vasorum* resides in the right sides of the heart and releases a complex mixture of proteins and proteases into the host's blood circulation. The parasite is in direct contact with host blood components and the endothelium of blood vessels. Over 1000 different excretory-secretory proteins (ESP) were recently identified to be released by this parasite *in vitro*. On its surface 1195 different proteins are present. Among ESP and surface proteins are many proteins relevant for nematode metabolism, but also putative modulators of host coagulation and more than 50 different proteases were identified [6]. Only few studies with a direct focus on coagulation or immune response upon *A. vasorum* infection have been conducted [3, 7–9] and the mechanisms and pathogenesis behind bleeding disorders induced by *A. vasorum* are still inconclusive. Prior to this work, we performed proteomic profiling of sera of *A. vasorum* experimentally infected dogs over time, where a decrease of proteins of the complement and coagulation cascade was identified [9]. We here questioned whether *A. vasorum* ESP modulate the host complement and coagulation cascade. We used *A. vasorum* ESP, containing proteases, and addressed their cleaving capacity in host blood by Terminal Amine Isotopic Labelling of Substrates (TAILS), a high-throughput quantitative proteomics method to distinguish protease-generated neo–N-termini from mature protein N-termini, to identify protease substrates and cleavage sites in complex biological samples such as blood [10, 11].

## Material and methods

### Collection of *A. vasorum* excretory/secretory proteins (ESP)

Live *A. vasorum* females were collected from the hearts and pulmonary arteries of 15 foxes hunted in the canton of Zurich, Switzerland. The hearts and pulmonary arteries were opened with sterile surgical scissors and adult specimens collected in sterile Petri dishes with fox blood and revitalised at 37°C. Adults were washed 3 x with warm sterile phosphate-buffered saline (PBS) and their viability checked. Live females were incubated for 1 h at 37°C and 5% $CO_2$ in 500 ml RPMI 1640 medium (Gibco, Thermo Fisher Scientific), supplemented with antibiotic-antimycotic solution (500 units penicillin, 0.5 mg streptomycin and 1.25 µg amphotericin B per ml, Thermo Fisher Scientific) and 50 µg/ml gentamicin solution (Sigma-Aldrich). They were then transferred to 500 ml fresh supplemented RPMI medium and incubated for 24 h at 37°C and 5% $CO_2$. Ninety adult females were kept in 30ml supplemented RPMI medium for 48 h under the same conditions. Medium containing *A. vasorum* ESP was then collected, centrifuged (1000 g, 10 min, 4°C) and supernatant filter-sterilized (0.22 µm). ESP were concentrated using a 10 kDa Amicon Ultra 15 ml centrifugal filter (Merck Millipore, US) and medium exchanged to PBS in the same device. Protein quantity was measured by Qubit protein assay (Thermo Fisher Scientific). ESP were stored at -80°C for 5 months until use.

### Dog serum and plasma collection and incubation with *A. vasorum* ESP

Serum and plasma were collected from 5 healthy male beagle dogs with haematology (complete blood count, including platelets) and coagulation (prothrombin time, partial thromboplastin time, and Clauss fibrinogen) values within reference ranges. Blood was drawn from the jugular vein into serum tubes (VACUETTE®, Greiner Bio-One) and 3.2% sodium citrate (1:9) coagulation tubes (VACUETTE®, Greiner Bio-One). The study was approved by the cantonal veterinary office, Zurich, Switzerland; animal trial permit no. 299776, 242/17. Thirty minutes after blood draw both serum and plasma tubes were centrifuged at 1800 g for 10 min (RT). Two hundred µl freshly collected serum and plasma from each dog (n = 5) was incubated for 6 h at 37°C with 30 µg female *A. vasorum* ESP (containing parasite-derived proteases). Serum and plasma control samples of each dog containing the same volume of PBS were incubated in parallel.

## Serum and plasma sample preparation

Immediately following incubation with ESP, samples containing ESP and control samples (5 replicates per group) were subjected to protein enrichment by ProteoMiner small capacity kit (Bio-Rad) according to the manufacturer's instructions, and further processed as previously described [10, 11]. Protein content was measured by Pierce™ BCA Protein Assay Kit (Thermo Fisher Scientific) before acetone/methanol precipitation. Briefly, 8 sample volumes of ice-cold acetone and 1 sample volume of ice-cold methanol were added to each sample and incubated for 2 h at -80°C. Samples were centrifuged (16,900 g, 15 min, 4°C), washed twice with ice-cold methanol and air-dried. Samples were then dissolved in 6 M guanidine hydrochloride (GuHCl) and adjusted to 2 M GuHCl and 200 mM HEPES (pH = 8). One hundred µg protein per sample were further processed. Samples were reduced with tris(2-carboxyethyl)phosphine (TCEP) at 10 mM and incubated for 30 min in a thermomixer at 37°C, 600 rpm, before alkylation with iodoacetamide (IAA) (25 mM final concentration), for 30 min in a thermomixer at 25°C, 500 rpm, in darkness. TMT labels (TMT10plex™ Label Reagent Set, Thermo Fisher Scientific) were used for N-terminal labelling. They were dissolved with one sample volume dimethyl sulfoxide (DMSO) and added in randomized order to the samples (0.8 mg label per 100 µg sample). Samples were incubated for 1 h at 25°C in darkness at 500 rpm. Twenty-five µl 1M ethanolamine was added to each sample to quench TMT labels, and samples incubated for 30 min in a thermomixer at 37°C, 500 rpm. After that, labelled samples containing ESP treated and control samples (5 each) were combined into a single tube. Samples were precipitated by acetone/methanol precipitation as described above, dissolved in 6 M GuHCl and diluted tenfold in 100 mM HEPES (pH = 8). Samples were then digested overnight in a thermomixer (37°C, 600 rpm) using 20 µg sequencing grade modified trypsin (Promega, V5113) (1:5 ratio). After digestion, 20 µg peptide sample was removed for separate analysis (pre-polymer samples). The samples were then treated overnight (37°C, 400 rpm) with HPG ALD polymer and 20 mM NaBH$_3$CN to remove internal tryptic peptides with N-terminal alpha-amines for negative selection of labelled peptides. Samples were adjusted to 100 mM Tris-HCl and incubated at 37°C for 30 min (600 rpm) before ultrafiltration at 10,000 g using 30 kDa Amicon Ultra 0.5 centrifugal filters (Merck Millipore) for polymer separation. The previously removed pre-polymer samples and the post-polymer samples were acidified with 5% trifluoroacetic acid (TFA) and purified using C18 OMIX pipette tips (Agilent). Briefly, tips were washed and equilibrated with 100% methanol, 60% acetonitrile (ACN)/0.1% TFA, and 3% ACN/0.1% TFA before sample binding. Aspirated peptides were then washed with 3% ACN/0.1% TFA and eluted with 60% ACN/0.1% TFA. Samples were dried to completeness using a speed-vac and resuspended in 3% ACN/0.1% formic acid (FA).

## LC-MS analysis

Peptide samples were diluted in 3% ACN, 0.1% FA to 1 µg/µl and retention time normalization peptides (iRT, Biognosys) added (1:20). Pre- and post-polymer samples were analysed in triplicates on an Orbitrap Fusion Lumos Tribrid mass spectrometer (Thermo Fisher Scientific) operated in line with an Acquity UHPLC M-class system (Waters) with a nanoEase M/Z Symmetry C18 trap column (100 Å, 5 um, 180 um x 20 mm, Waters) and a nanoEase M/Z HSS C18 T3 analytical column (100 Å, 1.8 um, 75 um x 250 mm Column, Waters) in data dependent acquisition mode (DDA). Samples were separated on a linear gradient from 5% to 32% solution B (0.1% FA in ACN) at a constant flow of 300 nl/min. A 10 µm fused-silica spray tip emitter (New Objective, PN) combined with a nano electrospray ionization (ESI) source (Digital PicoView 565, O/N: DPV-550-565, New Objective, Woburn, MA) ionized the eluted peptides. MS1 scans ranged from 375–1500 m/z and were recorded in profile mode and positive

polarity. Orbitrap resolution was set to 120,000 with an automated gain control (AGC) target of 4e5, and maximum injection time (maxIT) of 50 ms. MS2 scans (DDA) were recorded in centroid mode and positive polarity with 50,000 orbitrap resolution. The AGC target was 1e5 and the maxIT 105 ms. Isolated precursors were fragmented with higher-energy collisional dissociation (HCD) at a collision energy of 38% in the orbitrap. DDA scans first mass was set to 100 m/z and covered an isolation window of 0.7 m/z with quadrupole isolation. Dynamic exclusion of ions from fragmentation was set to 15 s after first occurrence. Ion charge states of 2–5 were recorded.

Proteome Discoverer 2.4 (Thermo Fisher Scientific) was used for data analysis. The S/N threshold was set to 5 for peak filtering. MS2 spectra were matched to the proteomes of *Canis lupus familiaris* (UP000002254), *Angiostrongylus costaricensis* (UP000050601), and *Angiostrongylus cantonensis* (UP000035642) under the following settings: spectrum matching for b and y ions; precursor mass tolerance: 10 ppm; fragment mass tolerance: 0.02 Da; dynamic peptide modifications: oxidation (M), N-terminal TMT10plex or acetylation; dynamic protein modifications: N-terminal Met-loss; static modifications: carbamidomethyl (C), TMT10plex (K); maximum missed cleavage sites: 2. The percolator target FDR was set to 0.01 and the integration tolerance to 20 ppm. Reporter abundance was based on S/N with a co-isolation threshold of 50 and an average reporter S/N threshold of 3. Ion abundance was normalized by the total peptide amount.

The generated mass spectrometry data have been deposited to the ProteomeXchange Consortium via the PRIDE [12] partner repository (dataset identifier PXD056659).

## Data analysis

Abundance ratios (log2) of > 1 and adjusted p-values of ≤ 0.05, obtained through Proteome Discoverer, were considered proteolytic processes caused directly or indirectly by *A. vasorum* ESP. Protein and peptide sequence data was exported from Proteome Discoverer and further analysed. Human orthologue protein accessions and motif sequences (4 amino acids up- and downstream of the cleavage site: positions P4 to P4′) were retrieved from NCBI (blastp). Human motif orthologue sequences were further used to retrieve position specific cleavage information by TopFIND 4.0 [13]. Human protein orthologues from substrate candidates were also converted to gene symbols using DAVID (v. 6.8) [14] for gene set enrichment analysis by Enrichr [15] to obtain enriched pathways and gene ontology terms.

## Results

We performed N-terminome analysis by TAILS [10, 11]. Dog serum and plasma samples were treated with *A. vasorum* ESP to assess the capacity of *A. vasorum* ESP and proteases to cleave host blood proteins. Briefly, ESP treated and untreated serum and plasma samples were subjected to N-terminal labelling and further digested before removal of internal tryptic peptides for negative selection of labelled peptides. Peptides were analysed by LC-MS and cleaved substrates identified by quantitative comparison of treated to untreated samples (Fig 1A).

Plasma peptide spectrum matches (PSM) were assigned to 5339 specific peptide groups from the dog and/or *A. vasorum*: 5181 peptides matched canine proteins, 119 matched *A. vasorum*, and 39 were allocated to both species. Identified peptides were assigned to 430 and 69 dog and *A. vasorum* proteins, respectively. PSM from serum were allocated to 5892 peptide groups, with 5711 peptides assigned to the dog, 129 to *A. vasorum*, and 37 to both organisms. Specific peptides resulted in 475 and 84 dog and *A. vasorum* proteins, respectively. In plasma, 1477 N-terminal peptides were identified, comprising 1428 peptides of canine origin; 68 were acetylated at the N-terminus while 1360 showed N-terminal TMT labelling (Fig 1B). Serum

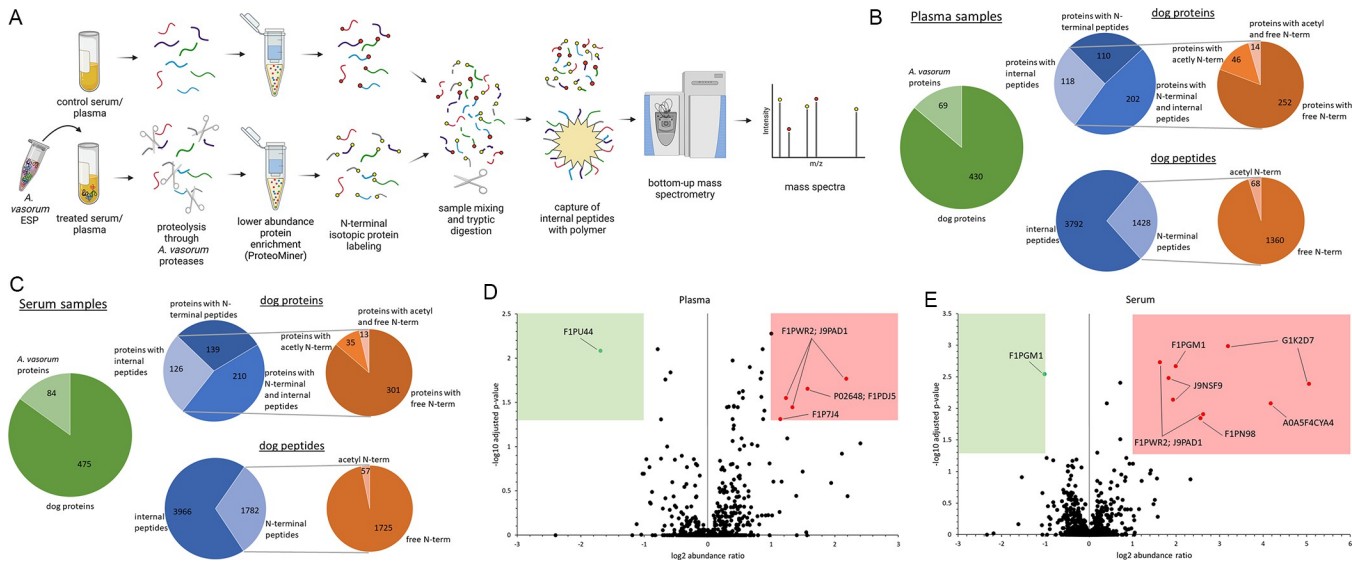

**Fig 1. Experimental approach and identified proteins.** A: Experimental approach by Terminal Amine Isotopic Labelling of Substrates (TAILS). This figure was created with BioRender.com. B: Identified dog and *Angiostrongylus vasorum* proteins, dog peptide groups, and N-terminal labelled peptides at protein and peptide level in dog plasma samples. C: Identified dog and *Angiostrongylus vasorum* proteins, dog peptide groups, and N-terminal labelled peptides at protein and peptide level in dog serum samples. D and E: Volcano plots of serum and plasma samples with significantly (log2 abundance ratio > 1 and adjusted p-value ≤ 0.05) cleaved proteins/peptides in red and proteins/peptides with log2 abundance ratio <- 1 and adjusted p-value ≤ 0.05 in green.

processing resulted in 1841 N-terminal peptides, among which 57 acetylated (N-terminal) dog serum peptides and 1725 N-terminal TMT labelled dog peptides were detected (Fig 1C). The remainder (n = 59) were N-terminally labelled *A. vasorum* peptides.

Normalized log2 abundance ratios were calculated for each peptide group to identify *A. vasorum* ESP induced cleavages. Among N-terminally labelled plasma dog peptides, 5 peptides were identified with log2 abundance ratio > 1 and adjusted p-value ≤ 0.05, originating from 3 different proteins, and one peptide with log2 abundance ratio < -1 and adjusted p-value ≤ 0.05 (Fig 1D). Among peptides obtained from dog serum samples, labelled at the N-terminal, 9 had a log2 abundance ratio > 1 and adjusted p-value ≤ 0.05, which were assigned to 6 different proteins (Fig 1E). One N-terminally labelled peptide with log2 abundance ratio <- 1 and adjusted p-value ≤ 0.05 was identified in serum samples (Table 1).

Eight proteins were significantly increasingly cleaved through *A. vasorum* ESP in serum and/or plasma, with one protein (F1PWR2; J9PAD1, complement C4-A) represented in both serum and plasma. Three proteins were coagulation factors (coagulation factors II, V and IX) and 3 were complement proteins (complement C3, C4-A and C5).

Further analysis with human motif sequence orthologues revealed 4 identical motifs in dogs and humans. Three human orthologous cleavage sites were identified as known cleavage sites. These were cleavage sites of coagulation factor IX (dog: G1K2D7 [position 369 and 370]; human: P00740 [position 371 and 372]) through thrombin (THRB) and cleavage of prothrombin (dog: J9NSF9 [position 324]; human: P00734 [position 325]) through kallikrein-4 (KLK4), coagulation factor X (FA10), mannan-binding lectin serine protease 2 (MASP2), or thrombin (Fig 2A).

Other identified cleavage positions in dog serum/plasma did not result in known human protease cleavage sites. A detailed list of all identified dog serum and plasma peptides passing our statistical significance cut-off, their motifs, human orthologue sequences and motifs, as well as known cleaving proteases in humans are presented in Table 1.

**Table 1. Proteins with identified cleavage sites.**

| Annotated Sequence | Sequence motif | Master Protein Accessions | Protein | Positions in Master Proteins | Abundance Ratios (log2) | Abundance Ratio Adj. P-Value | Sample type | Human orthologue | Human orthologue sequence | Human orthologue sequence motif | P1' Position human | Cleaving proteases |
|---|---|---|---|---|---|---|---|---|---|---|---|---|
| [D].KIAENGPFR.[I] | NNLD.KIAE | A0A5F4CYA4 | Lipocln_cytosolic_FA-bd_dom domain-containing protein | A0A5F4CYA4 [49–57] | 4.17 | 0.00825858 | Serum | Q8WX39 | RIKENGDLR | DDLN.RIKE | 49 | |
| [K].HLLPVTKPEIR.[S] | LQIK.HLLP | F1P7J4 | Complement C5 | F1P7J4 [755–765] | 1.14 | 0.04857235 | Plasma | P01031 | TLLPVSKPEIR | LHMK.TLLP | 756 | |
| [L].LLKDFDNVPPVVR.[W] | LALL.LLKD | F1PGM1 | Complement C3 | F1PGM1 [1243–1255] | 1.99 | 0.00213402 | Serum | P01024 | QLKDFDFVPPVVR | LALL.QLKD | 1242 | |
| [E].MKSSKPGWWLLNTEVGENQR.[A] | KTLE.MKSS | F1PN98 | Coagulation factor V | F1PN98 [1872–1891] | 2.56 | 0.01423608 | Serum | P12259 | MKASKPGWWLLNTEVGENQR | KTLE.MKAS | 1874 | |
| [K].AIHEKLGQYASPVAR.[R] | NFQK.AIHE | F1PWR2; J9PAD1 | Complement C4-A | F1PWR2 [681–695]; J9PAD1 [680–694] | 2.18 | 0.01694275 | Plasma | P0C0L4 | AINEKLGQYASPTAK | NFQK.AINE | 686 | |
| [F].YYHGDIPVANSLR.[V] | FVAF.YYHG | F1PWR2; J9PAD1 | Complement C4-A | F1PWR2 [541–553]; J9PAD1 [540–552] | 1.33 | 0.03566877 | Plasma | P0C0L4 | YYHGDHPVANSLR | FVAF.YYHG | 547 | |
| [Q].KAIHEKLGQYASPVAR.[R] | VNFQ.KAIH | F1PWR2; J9PAD1 | Complement C4-A | F1PWR2 [680–695]; J9PAD1 [679–694] | 1.23 | 0.02803427 | Plasma | P0C0L4 | KAINEKLGQYASPTAK | VNFQ.KAIN | 685 | |
| [N].FQKAIHEKLGQYASPVAR.[R] | RNVN.FQKA | F1PWR2; J9PAD1 | Complement C4-A | F1PWR2 [678–695]; J9PAD1 [677–694] | 2.62 | 0.01223316 | Serum | P0C0L4 | FQKAINEKLGQYASPTAK | RNVN.FQKA | 683 | |
| [K].AIHEKLGQYASPVAR.[R] | NFQK.AIHE | F1PWR2; J9PAD1 | Complement C4-A | F1PWR2 [681–695]; J9PAD1 [680–694] | 1.63 | 0.00183728 | Serum | P0C0L4 | AINEKLGQYASPTAK | NFQK.AINE | 686 | |
| [Q].YLKVPLVDR.[A] | SILQ.YLKV | G1K2D7 | Coagulation factor IX (Christmas factor) | G1K2D7 [369–377] | 5.05 | 0.00405378 | Serum | P00740 | YLRVPLVDR | LVLQ.YLRV | 371 | THRB |
| [Y].LKVPLVDR.[A] | ILQY.LKVP | G1K2D7 | Coagulation factor IX (Christmas factor) | G1K2D7 [370–377] | 3.19 | 0.00102313 | Serum | P00740 | LRVPLVDR | VLQY.LRVP | 372 | THRB |
| [E].LLCGASLISDR.[W] | SPQE.LLCG | J9NSF9 | Prothrombin (Coagulation factor II) | J9NSF9 [388–398] | 1.93 | 0.00725917 | Serum | P00734 | LLCGASLISDR | SPQE.LLCG | 389 | |
| [F].NEKTFGAGEADCGLRPLFEKR.[S] | QPFF.NEKT | J9NSF9 | Prothrombin (Coagulation factor II) | J9NSF9 [324–344] | 1.83 | 0.00327139 | Serum | P00734 | NPRTFGSGEADCGLRPLFEKK | QTFF.NPRT | 325 | KLK4; FA10; MASP2; THRB |
| [F].WDNLEKETEVLR.[Q] | TQEF.WDNL | P02648; F1PDJ5 | Apolipoprotein A-I | P02648 [95–106]; F1PDJ5 [95–106] | 1.57 | 0.02204595 | Plasma | P02647 | WDNLEKETEGLR | TQEF.WDNL | 96 | |
| [R].TLDPEAKGQEGVQR.[E] | LAIR.TLDP | F1PGM1 | Complement C3 | F1PGM1 [947–960] | -1.02 | 0.00283079 | Serum | P01024 | TLDPERLGREGVQK | VAVR.TLDP | 946 | |
| [S].LCPVDEAIHEKIQDDTSSLILETIR.[N] | CGKS.LCPV | F1PU44 | Resistin | F1PU44 [20–44] | -1.69 | 0.0081498 | Plasma | Q9HD89 | LCSMEEAINERIQEVAGSLIFRAIS | SSKT.LCSM | 21 | |

Dog serum and plasma proteins with identified cleavage sites induced through *Angiostrongylus vasorum* excretory/secretory proteins (including proteases) and their human orthologues, cleavage position orthologues, sequence motif orthologues, and known cleaving proteases.

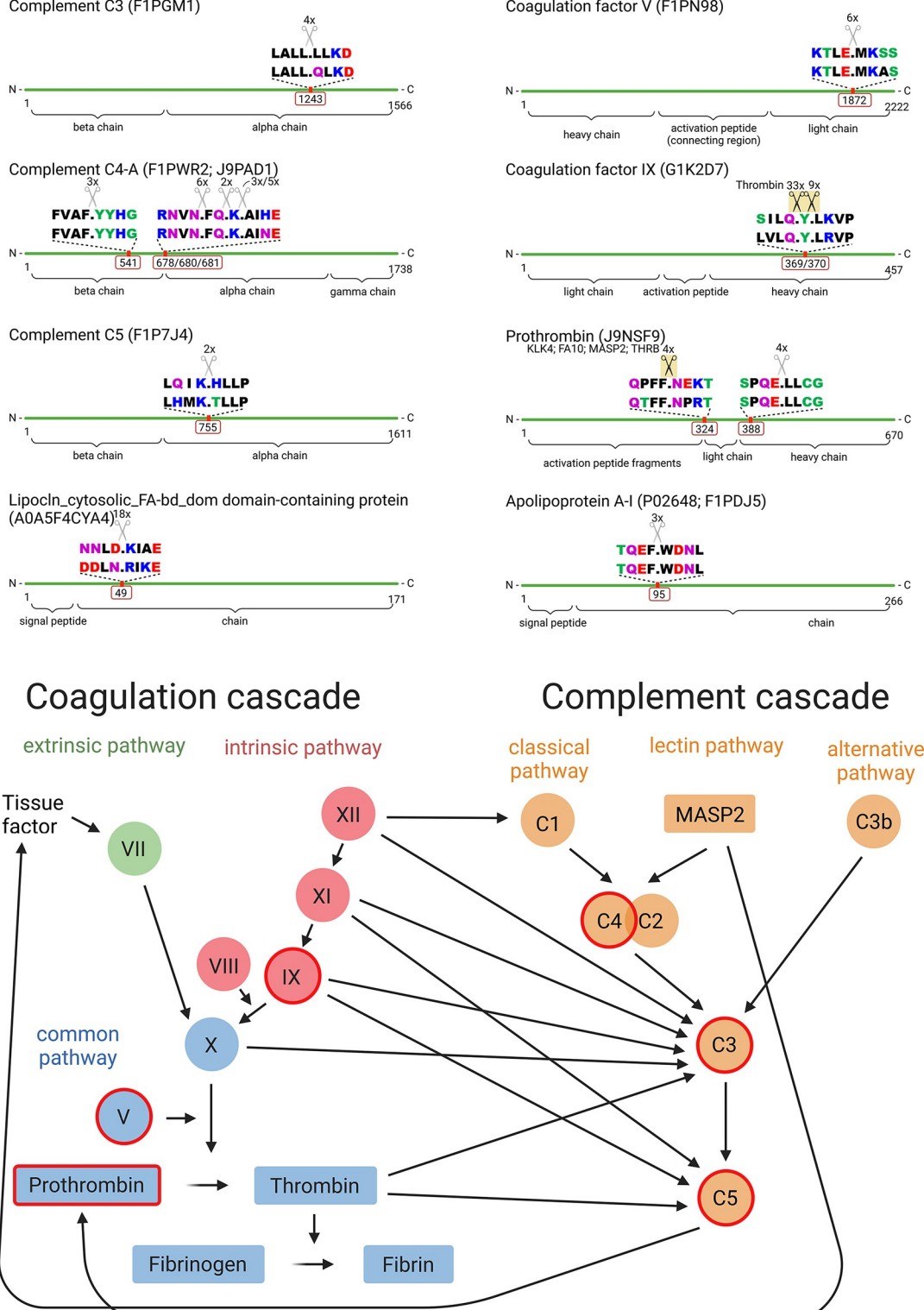

**Fig 2. Identified cleavage sites in proteins of the complement and coagulation cascade.** A: Cleavage sites of the 8 increasingly cleaved dog proteins through *Angiostrongylus vasorum* excretory/secretory proteins (including proteases) with comparison of the sequence motifs of dog (top sequence) and human (bottom sequence). The number in the red frame represents the amino acid sequence cleavage site. Known cleavage sites in humans are indicated with black scissors with a yellow background. Black: nonpolar, hydrophobic amino acids; red: polar, acidic amino acids; blue: polar, basic amino acids;

green: polar, neutral amino acids; purple: polar, neutral amino acids, amine group with dipole moment. KLK4: Kallikrein-related peptidase 4; FA10: Coagulation factor X; MASP2: Mannan binding lectin-associated serine protease-2; THRB: Thrombin. B: Simplified depiction of the mammalian complement and coagulation cascade and their interaction. Increasingly cleaved dog serum and plasma proteins through *Angiostrongylus vasorum* excretory/secretory proteins (including proteases) are framed in red. This figure was created with BioRender.com.

Gene enrichment analysis revealed several significantly enriched biological processes. Among them were regulation of complement activation (GO:0030449), regulation of humoral immune response (GO:0002920), and regulation of acute inflammatory response (GO:0002673). Among the top enriched Reactome pathways were 'Activation of C3 and C5', 'Regulation of complement cascade', and 'Formation of fibrin clot (clotting cascade)'. The top 10 significantly enriched biological processes and pathways are displayed in Table 2.

## Discussion

*Angiostrongylus vasorum* releases proteins and proteases into the host's blood circulation. The *A. vasorum* ESP have been characterized and several proteases have been identified, however,

**Table 2. Biological processes pathways.**

| Biological process | | | Reactome pathway | | |
|---|---|---|---|---|---|
| Term | Overlap | Adjusted P-value | Term | Overlap | Adjusted P-value |
| Cellular protein metabolic process (GO:0044267) | 6 of 484 | 1.15E-06 | Activation of C3 and C5 | 3 of 6 | 5.71E-08 |
| Regulation of protein activation cascade (GO:2000257) | 4 of 108 | 3.02E-06 | Regulation of complement cascade | 3 of 26 | 3.70E-06 |
| Regulation of complement activation (GO:0030449) | 4 of 109 | 3.02E-06 | Formation of fibrin clot (clotting cascade) | 3 of 39 | 8.64E-06 |
| Regulation of humoral immune response (GO:0002920) | 4 of 113 | 3.02E-06 | Complement cascade | 3 of 80 | 5.78E-05 |
| Regulation of immune effector process (GO:0002697) | 4 of 114 | 3.02E-06 | Transport of gamma-carboxylated protein precursors from the endoplasmic reticulum to the Golgi apparatus | 2 of 9 | 6.11E-05 |
| Regulation of acute inflammatory response (GO:0002673) | 4 of 121 | 3.20E-06 | Removal of aminoterminal propeptides from gamma-carboxylated proteins | 2 of 10 | 6.11E-05 |
| Regulation of protein processing (GO:0070613) | 4 of 128 | 3.44E-06 | Gamma-carboxylation of protein precursors | 2 of 10 | 6.11E-05 |
| Positive regulation of apoptotic cell clearance (GO:2000427) | 2 of 7 | 8.04E-05 | Gamma-carboxylation, transport, and amino-terminal cleavage of proteins | 2 of 11 | 6.53E-05 |
| Regulation of apoptotic cell clearance (GO:2000425) | 2 of 8 | 9.53E-05 | Common pathway of fibrin clot formation | 2 of 22 | 2.19E-04 |
| Post-translational protein modification (GO:0043687) | 4 of 357 | 1.40E-04 | Intrinsic pathway of fibrin clot formation | 2 of 22 | 2.19E-04 |

Top 10 biological processes and Reactome pathways obtained from increasingly cleaved dog serum and plasma proteins through *Angiostrongylus vasorum* excretory/secretory proteins (including proteases).

their activity and impact remain largely unexplored [6]. Proteolysis is a tightly regulated process, essential to various biological processes. In mammalian hosts, proteases usually circulate as inactive precursors, which are activated by cleavage processes that also include conformational change. Serine type proteases are of specific significance due to their diverse range of functions in e.g., coagulation and complement activation. In the coagulation cascade, serine proteases exhibit a preference for specific amino acid motifs upstream of the cleavage sites in their target proteins [16]. Parasitic helminths are known to express a variety of serine type proteases, which are needed for parasite development and feeding, and mediate interaction with the host [17]. For example, *Ancylostoma caninum*, a parasitic dog nematode, is known to release a proteolytic anticoagulant protease essential for its feeding [18]. Blood dwelling helminths are in constant contact with host blood and its coagulation components. Therefore, under homeostatic conditions, stable protease activity and balanced activation and suppression of coagulation favour parasite survival [19], whereas dysregulations may lead to complications, as observed in *A. vasorum* infected dogs with bleeding manifestations.

In the present study, we identified host proteins affected by *A. vasorum* ESP either directly or indirectly. Coagulation and complement proteins represented the most often targeted proteins by cleavage indicating an impact of *A. vasorum* on both the coagulation and complement cascade. Accordingly, decreased serum protein levels involved in the coagulation and complement pathways was previously observed in experimentally infected dogs upon *A. vasorum* infection [9].

Most of the identified cleavage sites were unknown and not in proximity to known protein processing sites. Coagulation factor V was cleaved in the light chain, which mediates binding and endocytosis of factor V by megakaryocytes [20]. Complement C3 was cleaved in its C3b alpha chain, more precisely in the C3d fragment, which is formed after the breakdown of C3b [21]. Finally, prothrombin was cleaved once in its heavy chain. These identified cleavage sites may suggest proteolysis of the mentioned proteins either directly or indirectly through *A. vasorum* ESP, leading to their inactivation or degradation. This is supported by the finding of reduced levels of coagulation factor V [9] and decreased factor V activity after experimental infection in dogs [5, 7]. In humans, coagulation factor V deficiency (congenital or acquired) can have a variable phenotypic expression. Moderate to severe bleeding disorders may be observed, with clinical signs of easy bruising and haematoma formation after trauma or medical treatment [22, 23]. There are phenotypes with low level factor V that do not show clinical signs of bleeding [24]. Acquired coagulation factor V deficiency leads to inhibition of factor V and is most seen after administration of chemical agents or drugs, and after surgical procedures. Other conditions that can lead to coagulation factor V deficiency in humans are cancer, infections, autoimmune disorders, blood transfusions, and organ transplantation [25]. Still in humans, inactivation or deficiency of complement C3 can lead to recurring bacterial infections and renal disease [26].

Two cleavage areas were identified within complement C4: one within its beta chain and several cleavage sites close to the propeptide cleavage area, which marks the N-terminal end of C4a anaphylatoxin (alpha chain) [27]. The identified cleavage site in complement C5 corresponds to a sequence within the C5 alpha chain, only four amino acids away from the cleavage site of the C5a anaphylatoxin C-terminus in humans. Human orthologous sequences were used to identify cleavage sites, as humans are the most closely related and well-annotated species compared to dogs. Dogs and humans however are rather far removed from each other, hence the precision of known cleavage sites needs to be questioned. The observed discrepancy of only few amino acids may be the result of sequential cleavages or trimming through endopeptidases [28]. The identified site within C5 may represent, for instance, the cleavage site that releases C5a anaphylatoxin from the C5 alpha chain in dogs. In contrast, the identified

cleavage site in C4 may provide evidence of sequential maturation, such as the removal of the propeptide. Only a few identified cleavage sites were identical between dogs and humans, impeding a direct comparison of identified cleavage areas. For instance, we observed one known cleavage site within coagulation factor IX for a human protease. Within this particular protein however, two cleavage sites were identified in close proximity. This is likely a result of truncation or trimming of one or several amino acids through exo- or endopeptidases [28]. We did not only observe this in coagulation factor IX, but also in complement C4, where 3 different cleavage sites around the propeptide cleavage area were identified.

Overall, only 3 cleavage sites in 2 proteins matched with known human orthologue sequence cleavage sites. The identified cleavage site in prothrombin matched with known cleavage in humans through thrombin and mannan-binding lectin serine protease 2 (Fig 2). The cleavage site is in proximity to the N-terminus of the light chain, which represents the activation site of prothrombin through coagulation factor X [29], suggesting possible activation of prothrombin to its active form thrombin, which in turn could lead to increased fibrin formation. Thrombin has however also been reported to induce proteolytic inactivity in coagulation factor IX [30], which we identified in the coagulation factor IX heavy chain in serum samples. This suggests that *A. vasorum* ESP have either directly or indirectly activated thrombin, which in turn likely led to the degradation of coagulation factor IX. In dogs and humans, deficiency of coagulation factor IX is known as haemophilia B. It manifests with spontaneous and joint bleeding in both dogs and people [31, 32]. Coagulation factor IX, once activated, serves in the activation of coagulation factor X together with coagulation factor VIII, via the intrinsic pathway of coagulation. We therefore hypothesise, although partially unknown and without known human protein processing sites, that also the other identified cleavage sites in dog proteins of proteins involved in the complement and coagulation cascade may contribute to the clinical picture of bleeding in *A. vasorum* infected dogs.

Apolipoprotein A-I is a high-density lipoprotein that plays a key role in cholesterol haemostasis. It contributes to the reverse transport of cholesterol from tissue to the liver and the excretion of cholesterol by efflux from tissue. It also has antioxidant, anti-inflammatory and antithrombotic properties [33]. Its increased cleavage may be a contributing factor to disease development in infected dogs. Lastly, lipocalin/cytosolic fatty acid binding domain-containing protein belongs to a group of fatty acid binding proteins that bind hydrophobic ligands and transport them through the cell. They can bind fatty acids, eicosanoids, and other cellular substrates [34].

Even though there are limited resources on dog proteome cleavage sites and *A. vasorum* protease specificity, the identified cleavage sites in the different coagulation and complement proteins suggest either activation or degradation of complement and coagulation components. This is also reflected in biological processes and pathways obtained by cleaved proteins: The biological processes, i.e., regulation of protein activation cascade (GO:2000257) and regulation of protein processing (GO:0070613), indicate that cleaved proteins are involved in protein processing and cascades. The pathways 'Complement cascade', 'Regulation of complement cascade', 'Activation of C3 and C5', 'Formation of fibrin clot (clotting cascade)', 'Common pathway of fibrin clot formation', and 'Intrinsic pathway of fibrin clot formation' clearly depict that the cleaved proteins are involved in the complement cascade and blood clotting.

We worked with both serum and plasma samples. Some coagulation factors, e.g., coagulation factor IX can be found in both plasma and serum [35]. Most other coagulation factors however are only present in plasma and may only be detected in trace amounts in serum, as they are usually bound in the blood clot formed for serum production. Increased cleavage of coagulation factor V, IX, and prothrombin was only observed in serum samples. Cleavage in plasma samples may have been inhibited due to the unavailability of calcium, which is bound

in blood samples collected with sodium citrate tubes used for plasma production. Therefore, calcium-dependent proteases would not have been able to function. This is likely the reason why we identified fewer cleavage sites in plasma samples, because most coagulation factors are calcium-dependent [36].

The applied TAILS technique has been used in the past to study proteases of parasites, bacteria, and viruses [37–40] and toxins [41]. Different medical conditions such as inflammatory disorders, cancer, and heart disease or treatment thereof have been investigated by TAILS [42–44]. Snake venom has been investigated using a N-terminomics approach since proteolysis is a major role in snake venom induced pathologies, which also includes bleeding disorders [41]. To the authors' knowledge this is the first study addressing a parasitic dog disease inducing coagulation by TAILS. TAILS is a powerful method that requires careful interpretation of the results due to potential biases related to labelling, substrate complexity, and data analysis. We acknowledge that our findings will need future validation in order to fully ascertain cleavage site identities.

Summarising, the current data demonstrate an interaction between *A. vasorum* ESP with the coagulation and complement cascade of the host and show that parasite-derived proteins can modulate host coagulation and complement proteins, with potential functional consequences on the host immune response and blood clotting. These findings further show that proteases present in *A. vasorum* ESP likely contribute to the pathogenesis of bleeding disorders–via direct catalytic activity or indirectly, facilitating cleavage by host proteins -, which are observed in *A. vasorum* infected dogs. Whether the *A. vasorum* ESP proteolytic activity on the eight candidate proteins was direct or indirect remains unresolved. To validate the specific activity of *A. vasorum* proteins and proteases on host coagulation and complement proteins and for the implementation of better patient management further studies are needed. Within *A. vasorum* ESP 57 proteases or proteasome subunits have been previously identified [6], the identification of their cleavage ability and specificity is desirable. To further study specific parasite proteases or proteins, single proteins/proteases could be isolated using antibody based pull-down methods, or recombinant parasite proteins or proteases could be produced, with the aim to obtain specific insights on cleavage capacity and specificity of *A. vasorum* ESP. To understand protease function, one may probe hexamers containing the predicted or expected cleavage site to quantify the protease catalytic efficiency, taking the site's context into account [45]. This can be assisted by bioinformatic approaches, which enable to link proteases and substrates based on structure [46]. Particularly cysteine proteases may be of interest, since several pathogen derived-cysteine proteases are known to activate complement C3, C4, and C5 or to degrade or cleave C3 and C5 [47]. Complement proteins have many functions, from wound healing to immune defences, and altered levels could have multiple physiological consequences. The overall health implications of such interactions with the complement system, however, remain unclear. It can only be speculated that a deficiency in the complement pathway may have additional consequences on susceptibility to certain pathogens and autoimmune conditions (PMID: 38406130). The specific combinations of missing (depleted) complement components—and their quantitative levels—are critical determinants, which would need to be thoroughly characterized. Typically, cleavage of C3, C4, and C5 is critical for phagocyte recruitment to the site of infection [48].

Several *A. vasorum* proteases represent specific candidates of interests that could be further studied, such as e.g., cathepsin B-like cysteine proteinases and serine carboxypeptidases, since most proteases in the coagulation cascade are serine proteases. Further candidates are T1 family proteases as well as proteasome proteins, which are associated with proteasome activity. Since proteasomes are relevant for proteolysis and protein degradation, they may also contribute to degradation of coagulation and complement proteins. Future studies will focus on

isolation or recombinant production of specific *A. vasorum* proteases and computational methods to shed light on specific mechanisms and pathways of *A. vasorum* induced coagulopathies in dogs and to develop potential management strategies for infected animals.

## Acknowledgments

We would like to thank Sina Hasler for her help with laboratory procedures.

## Author Contributions

**Conceptualization:** Nina Germitsch, Tobias Kockmann, Lucienne Tritten.

**Data curation:** Nina Germitsch, Tobias Kockmann, Lucienne Tritten.

**Formal analysis:** Nina Germitsch, Tobias Kockmann.

**Funding acquisition:** Manuela Schnyder.

**Investigation:** Nina Germitsch, Tobias Kockmann.

**Methodology:** Nina Germitsch, Tobias Kockmann, Lucienne Tritten.

**Project administration:** Manuela Schnyder, Lucienne Tritten.

**Resources:** Tobias Kockmann, Manuela Schnyder.

**Software:** Tobias Kockmann.

**Supervision:** Manuela Schnyder, Lucienne Tritten.

**Validation:** Nina Germitsch, Tobias Kockmann, Manuela Schnyder, Lucienne Tritten.

**Visualization:** Nina Germitsch.

**Writing – original draft:** Nina Germitsch.

**Writing – review & editing:** Tobias Kockmann, Manuela Schnyder, Lucienne Tritten.

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
