## [Decision Letter · Decision Letter 0]

7 Oct 2024

PONE-D-24-10029N-terminomics profiling of host proteins targeted by excretory-secretory proteases of the nematode Angiostrongylus vasorum identifies points of interaction with canine coagulation and complement cascadePLOS ONE

Dear Dr. Tritten,

Thank you for submitting your manuscript to PLOS ONE. After careful consideration, we feel that it has merit but does not fully meet PLOS ONE’s publication criteria as it currently stands. Therefore, we invite you to submit a revised version of the manuscript that addresses the points raised during the review process.

We look forward to receiving your revised manuscript.

Kind regards,

Elham Kazemirad, Ph.D

Academic Editor

PLOS ONE

Journal requirements: When submitting your revision, we need you to address these additional requirements. 1. Please ensure that your manuscript meets PLOS ONE's style requirements, including those for file naming. The PLOS ONE style templates can be found at https://journals.plos.org/plosone/s/file?id=wjVg/PLOSOne_formatting_sample_main_body.pdf and https://journals.plos.org/plosone/s/file?id=ba62/PLOSOne_formatting_sample_title_authors_affiliations.pdf 2. Please include a caption for figure 1A and 2A.

Reviewers' comments:

Reviewer's Responses to Questions

**Comments to the Author**

1. Is the manuscript technically sound, and do the data support the conclusions?

Reviewer #1: Yes

Reviewer #2: Partly

Reviewer #3: Yes

2. Has the statistical analysis been performed appropriately and rigorously? 

Reviewer #1: N/A

Reviewer #2: I Don't Know

Reviewer #3: Yes

3. Have the authors made all data underlying the findings in their manuscript fully available?

Reviewer #1: Yes

Reviewer #2: No

Reviewer #3: Yes

4. Is the manuscript presented in an intelligible fashion and written in standard English?

Reviewer #1: Yes

Reviewer #2: Yes

Reviewer #3: Yes

5. Review Comments to the Author

Reviewer #1: Dear authors, the research presented is exceptional. And consequently, opens a very rich field for for further investigation (as you pointed out). I tried my best to find even a minute error. It is my privilege to address to you, for the first time in my reviewer carrier that I do not have any remarks. Congratulations!

Reviewer #2: Germitsch et al. investigate the proteolytic processing in the canine plasma and serum samples, which were subjected to the nematode Angiostrongylus vasorum; it might be informative to have a detailed overview of the proteolysis present in the canine blood. The study is straightforward and a continuation of similar proteomic efforts by the corresponding group – in general, I am aware of the problems working with non-classical (lab) animals and the sub-per quality of the corresponding databases. Taking this into account, I find the study interesting, but there are two core points, which the authors would need to address: First, they are applying a ProteoMiner depletion kit on plasma and serum samples and comparing the different sample matrices with each other. But there is no thorough characterization of the depletion by this kit – especially, as the authors are trying to compare plasma to serum differences in proteolysis. But what is the alteration in the proteome composition of plasma and serum before/after the kit application? That is a very crucial question for this study and should be answered. Second, the mentioned PRIDE repository is not accessible and logging in as reviewer_pxd042869@ebi.ac.uk I get an error: “Invalid Accession or Token!”. I would like to check the deposited raw data/analysis files from Proteome Discoverer to follow the authors’ data analysis strategy.

Besides these two main points: the discussion about the results is a bit limited; especially after L352, what would be potential ways to apply the information gained by altered proteolysis in these dogs? How could you make use of this? Did you identify any cleavage which you could inhibit/promote to improve the health outcomes?

Some detailed comments:

- L57: Which timeframe was used in these experiments?

- L61: TAILS is far from being new; are there other methods, which might be used in a similar manner. Why did the authors pick TAILS – what is the rationale for this kind of workflow: maybe explain more, why such an approach might be useful?

- L97: Why was this kind of depletion used – is there any kind of justification? Where is any assessment of the kit effect? The authors should clearly show the alterations introduced by the kit, at least on the proteome-level by comparing the proteomes before/after depletion? Especially important for this study, as plasma and serum are compared – but are plasma or serum appropriately depleted to the same extent or do you introduce a bias merely by the depletion kit?

- L107: TMT6plex here and then TMT10plex in other places throughout the text? What exactly was the labeling strategy (6 or 10plex)? It should be mentioned correctly in the methods.

- L116: What was the applied protease-to-proteome ratio during digestion?

- L149: Why two strains and not only one – at least for me this is not obvious.

- L158: Invalid accession or token error in PRIDE – please correct.

- L162: Which adjustment (post-hoc correction) strategy was applied and is this built-in from Proteome Discoverer or self-calculated? If so, with which packages?

- L190: Concentrate only on the peptides, not PSMs.

- L210: For this table, only take the log2 abundance ratio, that is more informative. The “Distance to...” columns from TopFINDer are rather irritating here, as they are deduced from the homologous human cleavages. Maybe mention this in the text but remove it from the table.

- L304: Or might be the consequence of sequential cleavages by endo/carboxypeptidases processing N-termini/C-termini further - but avoid calling it irrelevant. You comment a little later exactly on that.

- L352: But the proteolytic activity will be before blood drawing, so does not seem to link. Why does plasma show less cleavage...

- L380: Would this be easy to check with inhibition by E-64 or similar compounds?

- Figures:

- Figure 1B+C: Rather present bar charts for these comparisons - the pie charts do not really serve the information you want to convey here.

- In the volcano plots: please add gene symbols in addition to protein IDs, as this is much easier to follow for the normal reader.

- Figure 1D: And which log2FC do we see here? It is not presented in the figure itself; maybe embed this information in the axis titles.

Reviewer #3: Methods:

Sample Specificity:

Why were both serum and plasma samples used in this study, and how could their distinct properties (e.g., calcium availability in plasma) influence the detection of cleavage sites and protease activity?

TAILS Method Application:

What potential biases or limitations does the TAILS (Terminal Amine Isotopic Labeling of Substrates) method introduce, especially regarding the identification of cleavage sites? How might these limitations affect the results?

Protease Identification:

Given that 57 proteases and proteasome subunits were identified in A. vasorum ESP, how were these proteases selected for further analysis, and how was their relevance to coagulation and complement pathways determined?

Host Protein Interactions:

How was the specificity of A. vasorum protease interaction with host proteins verified in this study, and how could cross-reactions with non-relevant host proteins be ruled out?

Results:

Cleavage Site Novelty:

Many of the identified cleavage sites in coagulation and complement proteins were novel. How confident can we be that these cleavage events are functionally relevant, given that they do not align with known human protein processing sites?

Coagulation Factor V and IX Cleavage:

Cleavage of coagulation factor V and factor IX was observed, which may suggest proteolytic degradation. How does this finding correlate with the clinical symptoms (e.g., bleeding disorders) seen in infected dogs, and are there any in vivo validations that directly link these cleavage events to the observed pathology?

Complement System Impact:

The study identified cleavage sites within complement proteins (C3, C4, C5). How do these specific cleavages influence the activation or inhibition of the complement pathway, and what are the potential downstream consequences for immune response in the host?

Discussion:

Mechanism of Proteolysis:

The study suggests that A. vasorum ESP can either directly or indirectly cleave host proteins involved in coagulation and complement pathways. What evidence supports the hypothesis of direct cleavage by parasite proteases, and how might indirect mechanisms (e.g., host protease activation) be distinguished?

Species-Specific Differences:

Only a few cleavage sites were conserved between dogs and humans. How could this affect the translatability of these findings to human parasitic infections or coagulation disorders, and what challenges arise in comparing dog and human orthologous proteins?

Future Directions for Protease Study:

The study highlights potential future directions, including isolating specific A. vasorum proteases for detailed analysis. What challenges do you anticipate in isolating and characterizing these proteases, particularly in verifying their cleavage specificity and relevance to the coagulation cascade?

6. PLOS authors have the option to publish the peer review history of their article (what does this mean?). If published, this will include your full peer review and any attached files.

Reviewer #1: **Yes: **Daniel Turudic

Reviewer #2: No

Reviewer #3: No

---

## [Author Response · Author response to Decision Letter 0]

5 Nov 2024

The title page and manuscript body have been adjusted according to the PLOS ONE guidelines.

2. Please include a caption for figure 1A and 2A.

Captions have been included for Figures 1 and 2.

The reference list has been reviewed. No retracted papers have been cited.

Reviewers' comments:

Reviewer's Responses to Questions 

Comments to the Author

1. Is the manuscript technically sound, and do the data support the conclusions?

Reviewer #1: Yes

Reviewer #2: Partly

Reviewer #3: Yes

2. Has the statistical analysis been performed appropriately and rigorously? 

Reviewer #1: N/A

Reviewer #2: I Don't Know

Reviewer #3: Yes

3. Have the authors made all data underlying the findings in their manuscript fully available?

Reviewer #1: Yes

Reviewer #2: No

Reviewer #3: Yes

4. Is the manuscript presented in an intelligible fashion and written in standard English?

Reviewer #1: Yes

Reviewer #2: Yes

Reviewer #3: Yes

5. Review Comments to the Author

Reviewer #1: Dear authors, the research presented is exceptional. And consequently, opens a very rich field for for further investigation (as you pointed out). I tried my best to find even a minute error. It is my privilege to address to you, for the first time in my reviewer carrier that I do not have any remarks. Congratulations!

Thank you very much, we are delighted with this very positive feedback, which appreciates our work and our conscientious and innovative approach.

Reviewer #2: Germitsch et al. investigate the proteolytic processing in the canine plasma and serum samples, which were subjected to the nematode Angiostrongylus vasorum; it might be informative to have a detailed overview of the proteolysis present in the canine blood. The study is straightforward and a continuation of similar proteomic efforts by the corresponding group – in general, I am aware of the problems working with non-classical (lab) animals and the sub-per quality of the corresponding databases. Taking this into account, I find the study interesting, but there are two core points, which the authors would need to address: First, they are applying a ProteoMiner depletion kit on plasma and serum samples and comparing the different sample matrices with each other. But there is no thorough characterization of the depletion by this kit – especially, as the authors are trying to compare plasma to serum differences in proteolysis. But what is the alteration in the proteome composition of plasma and serum before/after the kit application? That is a very crucial question for this study and should be answered. Second, the mentioned PRIDE repository is not accessible and logging in as reviewer_pxd042869@ebi.ac.uk I get an error: “Invalid Accession or Token!”. I would like to check the deposited raw data/analysis files from Proteome Discoverer to follow the authors’ data analysis strategy.

The discovery of clinically relevant biomarkers using modern proteomics has proven extremely challenging, primarily because of the large dynamic range of protein abundances in biofluids such as blood-derived matrices (e.g., plasma and serum), and the fact that only a small number of proteins constitute the vast majority of total blood protein mass. ProteoMiner columns contain a large, highly diverse bead-based library of combinatorial peptide ligands, which dilute high-abundance proteins and concentrate low-abundance proteins. High-abundance proteins in serum AND plasma samples (e.g., albumin) were saturated at high-affinity ligand(s) and excess protein washed away. Low-abundance proteins are concentrated on their specific affinity ligand(s). ProteoMiner can be used for differential expression analysis and is advertised for this use. Several papers highlighting the usefulness of these methods and confirm a high reproducibility and a low variance across replicates. The ProteoMiner kit is one of the most performant approach to maximize the total number of proteins detected by LC-MS/MS, much higher than what can be achieved after depletion of the most abundant proteins (https://doi.org/10.3390/ijms21165903). Reducing the dynamic range of proteins found in a complex sample represents the only way to detected low abundance proteins, which are certainly biologically relevant as well.

Data accession in PRIDE has been updated. PXD056659, Username: reviewer_pxd056659@ebi.ac.uk, Password: NgzAFslyBEti

Please use this link to access the login page: https://www.ebi.ac.uk/pride/login

Use the login on the left, using the reviewer login and password.

Besides these two main points: the discussion about the results is a bit limited; especially after L352, what would be potential ways to apply the information gained by altered proteolysis in these dogs? How could you make use of this? Did you identify any cleavage which you could inhibit/promote to improve the health outcomes?

Thank you. Since complement components have multiple functions, this is for now highly speculative. We now refer to a review listing pathologies associated with altered levels – in humans. We did not specifically search for cleavage sites that would compensate for the lacking complement components in this infection context. A quantitative dimension of the observed cleavages as well as a confirmation of the canine site identities will be the first next step. Ultimately, the goal would be of course to counter these effects, but this was out of the scope of our study. Please see L395, L400, and L414 for added discussion context.

Some detailed comments:

- L57: Which timeframe was used in these experiments?

In this previous study 3 serum samples per experimentally infected animal were tested. One sample prior to infection and 2 samples post infection (day 34 and day 75 post infection). Two individuals were sampled until day 220 post-inoculation. For further information on this study, please find full open access article here: https://www.nature.com/articles/s41598-020-79459-9

- L61: TAILS is far from being new; are there other methods, which might be used in a similar manner. Why did the authors pick TAILS – what is the rationale for this kind of workflow: maybe explain more, why such an approach might be useful?

Both N- and C-terminomics strategies can help elucidate protease substrates and cleavage sites, thereby the function of proteases in a given setting. TAILS has become an established protocol that has been implemented in the degradomics community with great success. N-terminomics analysis via TAILS represents a relatively easy and affordable approach, with the advantage that in this method, the polymer is water soluble and no unspecific binding of peptides occurs to the polymer. There are indeed multiple alternatives, and the choice of the strategy generally remains highly contextual. It is a powerful tool for identifying protease cleavage sites across the proteome. For broad discovery of cleavage events, TAILS/N-terminomics are highly recommended, while alternatives (e.g., PICS) are ideal for more targeted studies of protease specificity. Other techniques are appropriate for broad and specific discovery as well, often coming with a higher cost and/or time expense.

- L97: Why was this kind of depletion used – is there any kind of justification? Where is any assessment of the kit effect? The authors should clearly show the alterations introduced by the kit, at least on the proteome-level by comparing the proteomes before/after depletion? Especially important for this study, as plasma and serum are compared – but are plasma or serum appropriately depleted to the same extent or do you introduce a bias merely by the depletion kit?

Please see answer above regarding ProteoMiner. There are roughly 10 highly abundant proteins in blood (and also in serum and plasma, with the exception of fibrinogen, almost absent in serum). These 10 proteins account for 70-90% of the total protein contents and include albumin, immunoglobulins, fibrinogen, transferrin, haptoglobin, complement proteins, lipoproteins, alpha-1 antitrypsin, ceruloplasmin, C-reactive protein. But > 500 proteins are present in blood and blood products and are also physiologically relevant.

- L107: TMT6plex here and then TMT10plex in other places throughout the text? What exactly was the labeling strategy (6 or 10plex)? It should be mentioned correctly in the methods.

We thank the reviewer for pointing this out. It was corrected to 10plex throughout the manuscript.

- L116: What was the applied protease-to-proteome ratio during digestion?

The ratio was 1:5, this was added in the manuscript (L117).

- L149: Why two strains and not only one – at least for me this is not obvious.

Unfortunately, the Angiostrongylus vasorum proteome is not available. To have sufficient match we opted to use the Angiostrongylus proteomes that were available, which were A. cantonensis and A. costaricensis. 

- L158: Invalid accession or token error in PRIDE – please correct.

Data accession in PRIDE has been updated. PXD056659, Username: reviewer_pxd056659@ebi.ac.uk, Password: NgzAFslyBEti

- L162: Which adjustment (post-hoc correction) strategy was applied and is this built-in from Proteome Discoverer or self-calculated? If so, with which packages?

The PD consensus workflow was executed with built in post-hoc correction for adjusted p-values (PD version 2.4). This was clarified in the manuscript (L163).

- L190: Concentrate only on the peptides, not PSMs.

The PSM results were deleted from the results section.

- L210: For this table, only take the log2 abundance ratio, that is more informative. The “Distance to...” columns from TopFINDer are rather irritating here, as they are deduced from the homologous human cleavages. Maybe mention this in the text but remove it from the table.

The Abundance ratio column and the ‘Distance to’ columns have been deleted from the table. 

- L304: Or might be the consequence of sequential cleavages by endo/carboxypeptidases processing N-termini/C-termini further - but avoid calling it irrelevant. You comment a little later exactly on that.

This sentence was changed to reflect the statement in the next paragraph as suggested by the reviewer (L311).

- L352: But the proteolytic activity will be before blood drawing, so does not seem to link. Why does plasma show less cleavage...

Proteolytic activity in blood is activated after blood draw, this is how blood without addition of anticoagulants clots outside of the body. Hence, serum, in which clotting has fully occurred and proteolysis was activated, contains no or less coagulation factors as they are bound in the clotted part of the blood that is not used for analysis. Plasma is obtained after whole blood is collected into blood tubes that contain sodium citrate. The sodium citrate in these ‘plasma tubes’ inhibits the activation of the coagulation cascade by binding calcium which is needed for protease activity. Whole blood in ‘plasma tubes’ does not clot. The plasma component is then separated from the cells and collected, however the sodium citrate is mixed in with the plasma still and calcium still bound, which could inhibit activation of further calcium-dependent proteases present or added to plasma samples. 

This is mentioned in this discussion paragraph (L360-L369). We discuss that plasma likely showed less cleavage due to the unavailability of calcium. This section was adjusted to clarify why coagulation factors are not present or only present in trace amounts in serum (L363).

- L380: Would this be easy to check with inhibition by E-64 or similar compounds?

E-64 could be used to study A. vasorum derived cysteine proteases. However, for this publication it is out of scope as fresh ESP would have to be collected, specific proteases would have to be isolated and inhibition through E-64 would have to be evaluated. It, however, could be used in follow-up experiments. 

- Figures:

- Figure 1B+C: Rather present bar charts for these comparisons - the pie charts do not really serve the information you want to convey here.

We appreciate the reviewer’s comment but decided to keep the pie charts, as this form of visualization has been used by the authors in the past (https://doi.org/10.7554/eLife.27480). We think that the visualization of the proportions is important, and better conveyed by the current graph format. We have, however, adjusted the charts slightly to improve them.

- In the volcano plots: please add gene symbols in addition to protein IDs, as this is much easier to follow for the normal reader.

The available gene symbols have been added to the graphs.

- Figure 1D: And which log2FC do we see here? It is not presented in the figure itself; maybe embed this information in the axis titles.

This information can be found in the figure legend.

Reviewer #3: Methods:

Sample Specificity:

Why were both serum and plasma samples used in this study, and how could their distinct properties (e.g., calcium availability in plasma) influence the detection of cleavage sites and protease activity?

Whole blood could not be used for this study, so we opted to use both serum and plasma. Plasma contains coagulation components, whereas serum in theory does not. Serum however contains trace amounts. To ensure we can study accessible coagulation components we decided to use both serum and plasma as the sodium citrate may influence protease activity. How calcium availability may influence protease activity and cleavage is discussed in the discussion (L360-L369). The rationale behind working with serum here is to build on some of our previous work, performed on dog and fox banked sera (doi: 10.3390/pathogens10111513, doi: 10.3389/fcimb.2021.753320, doi: 10.1038/s41598-020-79459-9).

TAILS Method Application:

What 

---

## [Decision Letter · Decision Letter 1]

9 Dec 2024

N-terminomics profiling of host proteins targeted by excretory-secretory proteases of the nematode Angiostrongylus vasorum identifies points of interaction with canine coagulation and complement cascade

PONE-D-24-10029R1

Dear Dr. Tritten,

We’re pleased to inform you that your manuscript has been judged scientifically suitable for publication and will be formally accepted for publication once it meets all outstanding technical requirements.

Kind regards,

Elham Kazemirad, Ph.D

Academic Editor

PLOS ONE

Reviewers' comments:

Reviewer's Responses to Questions

**Comments to the Author**

1. If the authors have adequately addressed your comments raised in a previous round of review and you feel that this manuscript is now acceptable for publication, you may indicate that here to bypass the “Comments to the Author” section, enter your conflict of interest statement in the “Confidential to Editor” section, and submit your "Accept" recommendation.

Reviewer #2: All comments have been addressed

Reviewer #3: All comments have been addressed

2. Is the manuscript technically sound, and do the data support the conclusions?

Reviewer #2: Yes

Reviewer #3: Yes

3. Has the statistical analysis been performed appropriately and rigorously? 

Reviewer #2: Yes

Reviewer #3: Yes

4. Have the authors made all data underlying the findings in their manuscript fully available?

Reviewer #2: Yes

Reviewer #3: Yes

5. Is the manuscript presented in an intelligible fashion and written in standard English?

Reviewer #2: Yes

Reviewer #3: Yes

6. Review Comments to the Author

Reviewer #2: The authors addressed my comments and clarified my open questions. Especially the access to the PRIDE repository for the raw data/analysis is possible now.

Reviewer #3: (No Response)

7. PLOS authors have the option to publish the peer review history of their article (what does this mean?). If published, this will include your full peer review and any attached files.

Reviewer #2: No

Reviewer #3: **Yes: **Abolfazl Mirzadeh

---

## [Editor Report · Acceptance letter]

3 Jan 2025

PONE-D-24-10029R1 

PLOS ONE

Dear Dr. Tritten, 

I'm pleased to inform you that your manuscript has been deemed suitable for publication in PLOS ONE. Congratulations! Your manuscript is now being handed over to our production team.

Kind regards, 

on behalf of

Dr. Elham Kazemirad 

Academic Editor

PLOS ONE